# Intermediate Fine-Tuning Improves Mathematical Reasoning in Smaller Models

**Neeraj Gangwar**     **Suma P Bhat**     **Nickvash Kani**
Electrical and Computer Engineering
University of Illinois Urbana-Champaign, IL, USA
{gangwar2,spbhat2,kani}@illinois.edu

## Abstract

While large models pre-trained on high-quality data exhibit excellent performance across various reasoning tasks, including mathematical reasoning (e.g. GSM8k, MultiArith), specializing smaller models in mathematical reasoning remains a challenging problem. A common research approach to address this challenge involves distilling knowledge from large pre-trained teacher models into smaller student models. Other techniques include augmenting datasets by rephrasing questions or using multiple views of solutions to improve reasoning performance. In this work, we explore intermediate fine-tuning and show that fine-tuning a model on an arithmetic dataset before fine-tuning it on a reasoning dataset helps improve the model's performance on the reasoning tasks. The arithmetic dataset can be generated programmatically, eliminating the resource-intensive task of dataset creation. We evaluate the impact of intermediate fine-tuning using the original GSM8k training set and an expanded GSM8k training set created through distillation. Our experiments on multiple datasets demonstrate that intermediate fine-tuning leads to average improvements of 6.3% and 14.2% in reasoning tasks using the original and distilled training sets, respectively, with greedy decoding compared to the models fine-tuned directly on these sets.

## 1   Introduction

Scaling the model and data sizes has had a tremendous effect on performance across various natural language processing (NLP) tasks (Chowdhery et al., 2023; Achiam et al., 2023; Touvron et al., 2023b; Jiang et al., 2023). These pre-trained models can learn from a few demonstrations using in-context learning and do not require task-specific fine-tuning (Brown et al., 2020). They also benefit from generating a sequence of reasoning steps before arriving at the final answer. These strategies have been particularly effective for mathematical reasoning (Wei et al., 2022c; Nye et al., 2022; Fu et al., 2022; Zhou et al., 2022).

While these large models exhibit excellent performance on mathematical reasoning tasks, adapting smaller models for these tasks remains an open problem (Wei et al., 2022b). This problem is challenging because the math reasoning datasets, like GSM8k (Brown et al., 2020), consist of a small number of reasoning problems, generally accompanied by one solution. These datasets do not contain sufficient training examples to capture the complexity of math reasoning. To overcome this issue, a widely explored research direction is to distill knowledge from large pre-trained teacher models into smaller student models. Some methods use questions from existing training datasets and use prompting to generate solutions for fine-tuning smaller models (Ho et al., 2023; Magister et al., 2023; Fu et al., 2023; Hsieh et al., 2023; Yue et al., 2024). Others use various techniques to rephrase the questions to create more examples (Yu et al., 2024) or multiple views of solutions (Liang et al., 2024) to achieve better reasoning performance.

38th Conference on Neural Information Processing Systems (NeurIPS 2024).

This work focuses on using synthetically generated datasets to improve reasoning performance. We explore if fine-tuning a model on a related dataset before the reasoning datasets helps improve the model's reasoning abilities. Specifically, *we programmatically generate a dataset with arithmetic tasks and fine-tune the models on this dataset before fine-tuning them on the reasoning datasets*. In transfer learning literature, this is referred to as intermediate fine-tuning (Vu et al., 2020) or supplementary training (Phang et al., 2018). Empirical observations using several mathematical datasets lead to the following key takeaways:

- Models fine-tuned on the arithmetic dataset before a reasoning dataset perform better than the ones directly fine-tuned on the reasoning dataset. The arithmetic dataset can be generated programmatically, eliminating the need for manual resources.

- Based on our observations with multiple datasets with varying mathematical reasoning tasks, we find that intermediate fine-tuning results in better out-of-domain generalization.

- We evaluate our approach on datasets with relatively small and sufficiently large training sets. Our experiments show that intermediate fine-tuning improves performance in both cases.

Our source code and datasets are publicly available.[1]

## 2 Related Work

Adapting a pre-trained model for a downstream task has been traditionally done through task-specific fine-tuning. However, this approach does not work for mathematical reasoning tasks because datasets, like GSM8k, do not contain enough examples to capture the complexity of mathematical reasoning. Several works have focused on distilling multi-step reasoning solutions from large teacher models to overcome this limitation. Fu et al. (2023) prompted Codex (Chen et al., 2021) to generate multiple multi-step solutions for the examples in the GSM8k training set and fine-tuned FlanT5 on the ones that led to the correct final answer. Hsieh et al. (2023) used PaLM-540B (Chowdhery et al., 2023) for generating solutions and fine-tuned T5 (Raffel et al., 2020) in a multi-task setting to generate the labels and rationale. Liu et al. (2023) used GPT-3.5-turbo to generate synthetic GSM8k-like examples. Yue et al. (2024) showed that a hybrid of chain-of-thought and program-of-thought solutions performed better than using either format individually and created math generalist models – MAmmoTH. In addition to using LLMs to generate more solutions, Yu et al. (2024) used LLM rephrasing and backward reasoning to augment questions and created a new dataset called MetaMathQA.

Transfer learning has played a pivotal role in NLP. Vu et al. (2020) studied the effect of intermediate fine-tuning on the model's performance on a target task. Training on large multi-task mixtures is also a common trend in NLP (Aribandi et al., 2022; Wei et al., 2022a; Chung et al., 2024). Another research direction explores identifying relevant examples for a given downstream task from a huge collection of datasets, like P3 (Sanh et al., 2021). These methods create embeddings for all examples of interest using hidden states (Ivison et al., 2023) or gradients (Xia et al., 2024). Given a task, a small subset of relevant examples are selected based on similarity. These methods have been mainly applied to data-efficient instruction-tuning.

## 3 Our Approach

In this work, we fine-tune a model on an intermediate task before specializing it in mathematical reasoning. This is also referred to as intermediate fine-tuning (defined below). In our approach, we use an arithmetic dataset for intermediate fine-tuning. The arithmetic dataset is generated programmatically, thus eliminating the resource-intensive task of dataset creation.

### 3.1 Intermediate Fine-Tuning

Fine-tuning a model on an intermediate task before a downstream task can improve the model's performance on the said downstream task (Phang et al., 2018; Vu et al., 2020). The downstream task is also referred to as the target task. This is called intermediate fine-tuning and can lead to successful

---

[1]https://github.com/neerajgangwar/reasoning-ift

Table 1: Accuracy (%) achieved by models fine-tuned on the GSM8k datasets w/ and w/o intermediate fine-tuning (IFT) on the arithmetic dataset. We report the accuracy values with greedy and consistency decoding, separated by "/", with the greedy accuracy on the left. Model performance on MultiArith, ASDiv, and SVAMP is included to demonstrate no loss in out-of-domain generalization.

| Training Dataset | Model | IFT | GSM8k | MultiArith | ASDiv | SVAMP |
|---|---|---|---|---|---|---|
| GSM8k (Orig.) | FlanT5-Base | ✗ | 7.7 / 9.1 | 17.2 / 17.4 | 8.5 / 8.6 | 6.6 / 7.7 |
| | | ✓ | 10.5 / 12.8 (+2.8 / +3.7) | 26.1 / 31.5 (+8.9 / +14.1) | 11.2 / 12.8 (+2.7 / +4.2) | 9.1 / 9.9 (+2.5 / +2.2) |
| | FlanT5-Large | ✗ | 12.9 / 14.7 | 28.9 / 29.1 | 15.7 / 16.8 | 12.1 / 12.6 |
| | | ✓ | 17.1 / 18.0 (+4.2 / +3.3) | 49.4 / 54.1 (+20.5 / +25.0) | 21.0 / 21.4 (+5.3 / +4.6) | 15.3 / 15.9 (+3.2 / +3.3) |
| | T5v1.1 LM Adapt | ✗ | 6.6 / 6.7 | 13.3 / 12.8 | 4.9 / 6.4 | 5.7 / 6.1 |
| | | ✓ | 6.6 / 8.7 (0.0 / +2.0) | 22.8 / 24.1 (+9.5 / +11.3) | 10.3 / 11.4 (+5.4 / +5.0) | 7.5 / 8.2 (+1.8 / +2.1) |
| GSM8k (Dist.) | FlanT5-Base | ✗ | 17.5 / 19.9 | 31.1 / 33.9 | 23.6 / 24.6 | 20.4 / 20.2 |
| | | ✓ | 21.4 / 25.0 (+3.9 / +5.1) | 65.0 / 68.5 (+33.9 / +34.6) | 34.8 / 36.3 (+11.2 / +11.7) | 26.9 / 29.8 (+6.5 / +9.6) |
| | FlanT5-Large | ✗ | 22.4 / 24.9 | 45.0 / 43.7 | 29.1 / 30.2 | 23.2 / 25.0 |
| | | ✓ | 27.7 / 30.5 (+5.3 / +5.6) | 74.4 / 77.0 (+29.4 / +33.3) | 40.4 / 42.6 (+11.3 / +12.4) | 35.6 / 37.9 (+12.4 / +12.9) |
| | T5v1.1 LM Adapt | ✗ | 17.2 / 18.9 | 35.6 / 35.6 | 23.0 / 24.7 | 19.4 / 20.8 |
| | | ✓ | 22.6 / 24.0 (+5.4 / +5.1) | 49.4 / 54.6 (+13.8 / +19.0) | 30.7 / 31.9 (+7.7 / +7.2) | 24.1 / 24.8 (+4.7 / +4.0) |

knowledge transfer for similar intermediate and target tasks. Vu et al. (2020) have shown that this helps tasks with limited training examples and sufficiently large training sets. Building on these results, we explore if an intermediate task can be leveraged to improve the model's mathematical reasoning performance.

## 3.2 Intermediate Task for Math Reasoning

Following prior work on transfer learning (Vu et al., 2020; Neyshabur et al., 2020), while various mathematical datasets may help improve a model's reasoning performance, we focus on an arithmetic dataset in this work for two reasons. First, arithmetic computations are an integral part of mathematical reasoning. Second, while curating a math reasoning dataset requires considerable resources, a simple arithmetic dataset can be generated programmatically. We leave the exploration of other potential intermediate tasks for future work.

We refer to Liu and Low (2023) to programmatically generate an arithmetic dataset. Their work has shown that LLaMA (Touvron et al., 2023a) fine-tuned on a programmatically generated dataset outperforms GPT-4 (Achiam et al., 2023) on arithmetic tasks. While the dataset from Liu and Low (2023) contains the basic arithmetic operations – addition, subtraction, multiplication, and division, we also include fractions and percentages. GSM8k does not require computations over large numbers, hence we limit the number of digits in the operands to seven. Furthermore, we use log-uniform sampling to ensure that the dataset is not skewed towards numbers with greater digits. This dataset contains $\sim$ 1.3M examples.

## 3.3 Datasets

**Training.** We use GSM8k for model specialization. As it does not have a validation set, we randomly sample 512 examples from the training set to create a validation set. We use two versions of GSM8k in this work.

*Original.* In the first version, we use the remaining examples from the training set for model specialization. This dataset contains 6961 examples. We refer to this dataset as GSM8k (Orig.).

*Distilled.* We generate a distilled dataset using the questions from GSM8k (Orig.) to evaluate if intermediate fine-tuning benefits tasks with large training datasets. This dataset is generated by prompting Mistral-7B (Jiang et al., 2023) using the prompt from Wei et al. (2022c). We generate 64 solutions per question and keep the ones that lead to the correct final answer. After removing duplicate solutions, this results in a dataset with $\sim$175k examples. We refer to this dataset as GSM8k (Dist.).

**Evaluation** We use the GSM8k test set to evaluate the models. We also use three additional datasets – MultiArith (Roy and Roth, 2015), ASDiv (Miao et al., 2020), and SVAMP (Patel et al., 2021) – to test out-of-domain generalization. MultiArith contains problems focused on basic arithmetic operations and is relatively simpler than GSM8k. ASDiv focuses on diverse language usage patterns

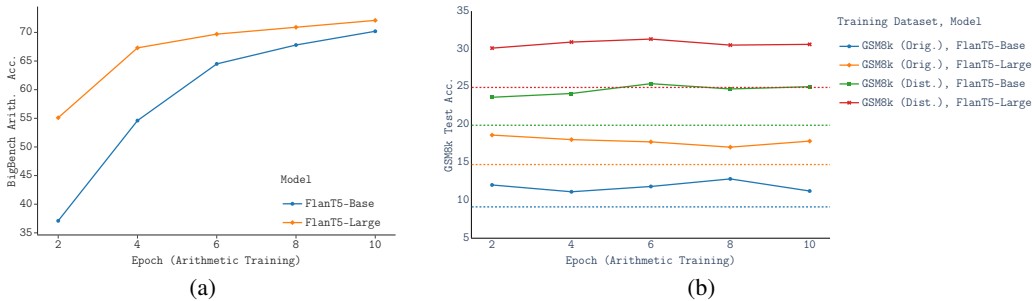

Figure 1: (a) Accuracy (%) on the BigBench Arithmetic benchmark at different intervals during the intermediate fine-tuning. (b) GSM8k test accuracy (%) of the models initialized from different checkpoints in the intermediate fine-tuning. The dotted lines of the same color correspond to the models directly fine-tuned on GSM8k.

and covers wide problem types taught in elementary school. SVAMP contains problems with varying structures to ensure that a model cannot solve the problems by applying simple heuristics and ignoring question text.

## 4  Experiments

### 4.1  Training Details

For our experiments, we use FlanT5 (Chung et al., 2024) which is an instruction-tuned T5 (Raffel et al., 2020). The base and large versions of FlanT5 are used with 250M and 750M parameters, respectively. We use the AdamW optimizer (Loshchilov and Hutter, 2017) with a learning rate of $10^{-4}$, a weight decay of $10^{-4}$, and an effective batch size of 128. For FlanT5-Large, a learning rate warmup of 500 steps is used. All model training processes are initialized using a fixed seed.

The intermediate fine-tuning is performed for 10 epochs without validation, and checkpoints are saved every other epoch. To adapt these models for reasoning, we continue the training from these checkpoints on GSM8k. The models are fine-tuned for 20 and 100 epochs on GSM8k (Dist.) and GSM8k (Orig.), respectively. The best checkpoint is selected based on the GSM8k validation performance.

We use greedy and consistency decoding at inference. For consistency decoding, nucleus sampling (Holtzman et al., 2019) is used with $T = 0.6$ and $p = 0.9$ to sample eight responses, and the most consistent final answer is chosen. As nucleus sampling is a stochastic decoding method, we repeat the evaluation with consistency decoding three times and report the mean accuracy.

### 4.2  Results

**In-Domain Performance.**  We first evaluate the models on the GSM8k test set. Table 1 shows the accuracy (%) achieved by different models. We observe that both FlanT5-Base and FlanT5-Large benefit from intermediate fine-tuning, and the performance on GSM8k improves significantly. These results also show that intermediate fine-tuning helps with both GSM8k (Orig.), which has a small training set, and GSM8k (Dist.), which already has a sufficiently large training set.

**Out-of-Domain Performance.**  Next, the models fine-tuned on GSM8k are evaluated on MultiArith, ASDiv, and SVAMP, and the results are shown in Table 1. These results indicate that intermediate fine-tuning does not harm out-of-domain generalization. The models fine-tuned on the arithmetic dataset first generalize better than those directly fine-tuned on GSM8k.

**Arithmetic vs GSM8k Performance.**  We use the checkpoints fine-tuned on the arithmetic dataset to initialize the models to be fine-tuned on GSM8k. But does good performance on arithmetic tasks always translate to better GSM8k performance? Our experiments show that this is not always the case. We use the BigBench Arithmetic benchmark (BIG-bench authors, 2023) to evaluate the model's

arithmetic abilities and report micro-averaged accuracy across addition, subtraction, multiplication, and division. The results of this experiment are shown in Figure 1. The models fine-tuned on the arithmetic dataset perform better on the BigBench Arithmetic benchmark as the training progresses (Figure 1a). However, the models initialized from the checkpoints with better arithmetic performance do not always result in better GSM8k performance (Figure 1b). These results agree with the findings of Neyshabur et al. (2020) which show, for image datasets, that pre-training performance is not always a faithful indicator of effective transfer learning. This makes it harder to decide when to stop the intermediate fine-tuning. However, it should be noted that all checkpoints from intermediate fine-tuning result in a better performance than directly fine-tuning the model on GSM8k.

## 5 Conclusion

In this work, we examined intermediate fine-tuning for mathematical reasoning. Our experiments showed that fine-tuning a model on a programmatically generated arithmetic dataset before a reasoning dataset helped improve the model's performance on the reasoning tasks. We evaluated our approach with small and large datasets, and intermediate fine-tuning resulted in better performance in both cases. Moreover, intermediate fine-tuning did not harm out-of-domain generalization, instead, models fine-tuned on the arithmetic dataset first showed better out-of-domain generalization. Finally, while this work does not offer a method for determining when to stop intermediate fine-tuning, initializing models from any checkpoints during the process yielded better results than fine-tuning them directly on GSM8k.

## 6 Limitations

While intermediate fine-tuning improved a model's mathematical reasoning performance, good performance on the arithmetic tasks did not always lead to better reasoning performance. Due to this, deciding the maximum number of epochs for intermediate fine-tuning and selecting a checkpoint from this process to initialize the next model remain open problems. Furthermore, identifying an intermediate task or programmatically generating a dataset for it may not always be feasible, limiting the applicability of this approach. Other mathematical datasets for intermediate fine-tuning may also be explored. Finally, we evaluated our approach on GSM8k. The experiments presented in this work may be extended to include other mathematical reasoning datasets, like MATH (Hendrycks et al., 2021). We leave these avenues for future research.

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
