# OpenReview forum: "Intermediate Fine-Tuning Improves Mathematical Reasoning in Smaller Models"
_NeurIPS.cc/2024/Workshop/MATH-AI — MATH-AI 24_

### Official Review · Reviewer_dCvS · 2024-10-02

**Rating:** 5
**Confidence:** 4

**Review:**

The paper introduces an effective method for enhancing mathematical reasoning in smaller models via intermediate fine-tuning. This resource-efficient and scalable approach yields significant performance improvements. However, the focus solely on addressing arithmetic deficiencies lacks novelty. Further exploration of alternative tasks, checkpoint selection, and broader applications could enhance the paper's impact.

**Strengths:**

The proposed method is straightforward, and the authors conduct extensive experiments on GSM8K math reasoning tasks to validate its effectiveness.

**Weaknesses:**

1.	*Lack of Novelty:* Previous works have already tackled the issue of LLMs' poor performance on arithmetic operations, employing tools like programmatic thinking, symbolic languages, and fine-tuning to address these challenges.
2.	*Checkpoint Selection Issues:* The paper notes that strong performance on arithmetic tasks doesn't necessarily lead to improved reasoning performance. The absence of a clear method for selecting the optimal checkpoint during intermediate fine-tuning limits the approach's consistency.
3.	*Dataset Dependence:* The method heavily relies on the GSM8K dataset and a programmatically generated arithmetic dataset. Extending this approach to other reasoning tasks outside mathematics may require significant modifications or different intermediate datasets, which the paper does not address.
4.	*Unexplored Model Architectures:* The study does not investigate whether the advantages of intermediate fine-tuning are confined to specific model architectures like FlanT5. It would be valuable to determine if this technique generalizes across various architectures.

---

### Official Review · Reviewer_Qkxe · 2024-10-04
**Review for paper 54: Intermediate Fine-Tuning Improves Mathematical Reasoning in Smaller Models**

**Rating:** 8
**Confidence:** 3

**Review:**

**Summary of contributions:**

In this study the authors explore the challenge to tailor smaller models specifically for mathematical reasoning. To address this challenge, they delve into intermediate fine-tuning, a process where a model is fine-tuned on an arithmetic dataset before further fine-tuning on a reasoning dataset. The authors report intermediate fine-tuning step significantly improves the model’s performance on reasoning tasks.

**Relevance and Clarity:**

Relevant to the guiding theme of the proposed workshop and the experimental results are explicitly mentioned and well explained for easy understanding.

**Originality:**

The authors come up with a novel methodology for intermediate fine-tuning which boosts the performance of the fine-tuned model on the final target task and at the same time preserves prior knowledge of the model which is a rare phenomenon in the world of fine-tuned models.

**Strengths**

1. *Improvement in reasoning tasks without loss of prior knowledge:*

The experiments involving intermediate fine-tuning reveal considerable improvements in performance on the final reasoning tasks for multiple diverse datasets thus proving the model’s robustness to varying reasoning tasks without loss of its prior knowledge.

2. *Generation of synthetic dataset:*

The authors’ endeavor to programmatically generate a dataset with arithmetic tasks overcomes the challenge of resource intensive and time-consuming dataset creation. Also, including fractions and percentages as well in addition to the basic arithmetic operations in the dataset is noted.

3. *Discussion of related work:*

All relevant works are discussed and properly cited within the main body of the paper, with clear relationship to submission.

**Weakness**

1. As already reported by the authors, this research does not identify a methodology to determine the optimal stopping point for intermediate fine-tuning to ensure maximum accuracy on the final target task.

2. The research could have addressed one significant concern of fine-tuning which is fine-tuning can inherit biases present in the training data. Perhaps a better clarity on how the generated data for intermediate fine-tuning was tested for potential biases will make this research more engaging.

**Significance**

 This paper demonstrates the potential of intermediate fine-tuning on programmatically generated arithmetic dataset as a game changer to increase the mathematical reasoning ability of models and can serve as an alternative solution to knowledge distillation from large language models to smaller versions to achieve the same goal.

---

### Official Review · Reviewer_mcss · 2024-10-04
**Finetuning on arithmetic data is an interesting idea, but why does better arithmetic not result in better gsm8k scores?**

**Rating:** 7
**Confidence:** 4

**Review:**

The paper proposes to do an intermediate finetuning step on synthetically generated arithmetic dataset and show that it results in better performance on gsm8k than finetuning on that data directly.

This is an interesting idea and the results in Table 1 demonstrate significant gains both in and out-of-domain. Yet Figure 1 is really puzzling to me. What does the model learn during the intermediate finetuning if doing arithmetic better doesn't correspond to better gsm8k accuracy? I feel that without studying this more, this work is not fully conclusive. E.g., is it important to use arithmetic dataset or can some other datasets be used to achieve the same effects (e.g. some general instruction following data)? Also, does intermediate finetuning results in a model that makes fewer arithmetic mistakes on gsm8k validation/test set? You could try to include the exact calculations from gsm8k validation questions into the training dataset and see if at least those are being solved 100% correctly, since they become part of the training data during intermediate finetuning stage. If that doesn't happen (as Figure 1 potentially indicates), then I feel this needs more analysis and perhaps the take-away message that "arithmetic intermediate finetuning is helpful" is somewhat misleading as "arithmetic" might not even be necessary here.

Other than that, the paper is well written and presents an interesting idea that improves accuracy and efficiency of finetuning small, specialized models on mathematical reasoning tasks.

---

### Official Review · Reviewer_Hyzi · 2024-10-05
**Arithmetic training improves math reasoning performance for SLMs**

**Rating:** 6
**Confidence:** 4

**Review:**

The authors investigate the intermediate fine-tuning(ift) for math reasoning, and perform ift on T5 models to illustrate its effectiveness.

Pros:

- The experiments show that T5 models using ift outperform T5 without ift, with a clear gap.
- IFT does not hurt other OOD test set, and model’s arithmetic performance show strong correlation with math reasoning.

Cons:

- the statement of `OOD` is not very suitable, given the fact that MultiArith, SVAMP, and Asdiv basically contains the same questions with fewer reasoning steps compared with GSM8K.
- it would be better if the author could make some clarification about how to generate ift data, and what is the difference compared with the mentioned “GOAT” method. The current version looks like authors different applies the same GOAT method to generate arithmetic datasets.
- make fair comparison w.r.t FLOPs: the current experiments only consider model’s performance with or without IFT, however, the FLOPs of two methods are quite different. It’s better to train the baseline more epochs to match the FLOPs used in IFT.

---

### Decision · Program_Chairs · 2024-10-09

Accept